# Decompositional Generation Process for Instance-Dependent Partial Label Learning

**Congyu Qiao, Ning Xu**\*, **Xin Geng**\*
School of Computer Science and Engineering, Southeast University, Nanjing 210096, China
{qiaocy, xning, xgeng}@seu.edu.cn

## Abstract

Partial label learning (PLL) is a typical weakly supervised learning problem, where each training example is associated with a set of candidate labels among which only one is true. Most existing PLL approaches assume that the incorrect labels in each training example are randomly picked as the candidate labels and model the generation process of the candidate labels in a simple way. However, these approaches usually do not perform as well as expected due to the fact that the generation process of the candidate labels is always instance-dependent. Therefore, it deserves to be modeled in a refined way. In this paper, we consider instance-dependent PLL and assume that the generation process of the candidate labels could decompose into two sequential parts, where the correct label emerges first in the mind of the annotator but then the incorrect labels related to the feature are also selected with the correct label as candidate labels due to uncertainty of labeling. Motivated by this consideration, we propose a novel PLL method that performs Maximum A Posterior (MAP) based on an explicitly modeled generation process of candidate labels via decomposed probability distribution models. Extensive experiments on manually corrupted benchmark datasets and real-world datasets validate the effectiveness of the proposed method. Source code is available at https://github.com/palm-ml/idgp.

## 1 Introduction

Partial label learning (PLL) aims to deal with the problem where each instance is provided with a set of candidate labels, only one of which is the correct label. The problem of learning from partial label examples naturally arises in a number of real-world scenarios such as web data mining Luo & Orabona (2010), multimedia content analysis Zeng et al. (2013); Chen et al. (2017), and ecoinformatics Liu & Dietterich (2012); Tang & Zhang (2017).

A number of methods have been proposed to improve the practical performance of PLL. Identification-based PLL approaches Jin & Ghahramani (2002); Nguyen & Caruana (2008); Liu & Dietterich (2012); Chen et al. (2014); Yu & Zhang (2016) regard the correct label as a latent variable and try to identify it. Average-based approaches Hüllermeier & Beringer (2006); Cour et al. (2011); Zhang & Yu (2015) treat all the candidate labels equally and average the modeling outputs as the prediction. In addition, risk-consistent methods Feng et al. (2020); Wen et al. (2021) and classifier-consistent methods Lv et al. (2020); Feng et al. (2020) are proposed for deep models. Furthermore, aimed at deep models, Wang et al. (2022) investigate contrastive representation learning, Zhang et al. (2021) adapt the class activation map, and Wu et al. (2022) revisit consistency regularization in PLL.

It is challenging to avoid overfitting on candidate labels, especially when the candidate labels depend on instances. Therefore, the previous methods assume that the candidate labels are instance-independent. Unfortunately, this often tends to be the case that the incorrect labels related to the feature are more likely to be picked as candidate label set for each instance. Recent work Xu et al. (2021) has also shown that the presence of instance-dependent PLL imposes additional challenges but is more realistic in practice than the instance-independent case.

---

\*Corresponding authors.

In this paper, we focus on the instance-dependent PLL via considering the essential generating process of candidate labels in PLL. To begin with, let us rethink meticulously how candidate labels arise in most manual annotation scenarios. When one annotates an instance, though the correct label has already emerged in the mind of the annotator first, the incorrect labels which are related to the feature of the instance confuse the annotator, then leading to the result that the correct label and some incorrect labels are packed together as the candidate labels. Therefore, the generating process of the candidate labels in instance-dependent PLL could be decomposed into two stages, i.e., the generation of the correct label of the instance and the generation of the incorrect labels related to the instance, which could be described by Categorical distribution and Bernoulli distribution, respectively.

Motivated by the above consideration, we propose a novel PLL method named IDGP, i.e., Instance-dependent partial label learning via Decompositional Generation Process. Before performing IDGP, the distributions of the correct label and the incorrect label given the training example should be modeled explicitly by decoupled probability distributions Categorical distribution and Bernoulli distribution. Then we perform Maximum A Posterior (MAP) estimation on the PLL training dataset to deduce a risk minimizer. To optimize the risk minimizer, Dirichlet distribution and Beta distribution are leveraged to model the condition prior inside and estimate the parameters of Categorical distribution and Bernoulli distribution due to the conjugacy. Finally, we refine prior information by updating the parameters of the corresponding conjugate distributions iteratively to improve the performance of the predictive model in each epoch. Our contributions can be summarized as follows:

- We for the first time explicitly model the generation process of candidate labels in instance-dependent PLL. The entire generating process is decomposed into the generation of the correct label of the instance and the generation of the incorrect labels, which could be described by Categorical distribution and Bernoulli distribution, respectively.
- We optimize the models of Categorical distribution and Bernoulli distribution via the MAP technique, where the corresponding conjugate distributions, i.e., Dirichlet distribution and Beta distribution are induced.
- We derive an estimation error bound of our approach, which demonstrates that the empirical risk minimizer would approximately converge to the optimal risk minimizer as the number of training data grows to infinity.

## 2 RELATED WORK

In this section, we briefly review the literature for PLL from two aspects, i.e., traditional PLL and deep PLL. The former absorbs many classical machine learning techniques and usually utilizes linear models while the latter embraces deep learning and builds upon deep neural networks. We focus on the underlying assumptions on the generation of candidate labels behind part of them.

Traditional PLL usually uses average-based or identification-based approaches to disambiguate the candidate label set. Average-based approaches treat each candidate label of the instance equally Hüllermeier & Beringer (2006); Cour et al. (2011); Zhang & Yu (2015). Typically, Hüllermeier & Beringer (2006); Zhang & Yu (2015) apply the K-nearest neighbor technique to predict a new instance through voting. Identification-based approaches are constantly trying to identify the possible correct label, either explicitly or implicitly during the training phase, in order to reduce label ambiguity Jin & Ghahramani (2002); Nguyen & Caruana (2008); Liu & Dietterich (2012); Chen et al. (2014); Yu & Zhang (2016). For instance, Nguyen & Caruana (2008); Yu & Zhang (2016) formulate their objective functions by treating the correct label as a latent variable via the maximum margin criterion. Zhang et al. (2016); Feng & An (2018); Wang et al. (2019); Xu et al. (2019b) use topological information in the feature space to iteratively update the confidence of each candidate label or the label distribution. It should be noted that most traditional PLL methods ignore the process of generating candidate labels. The Logistic Stick-Breaking Conditional Multinomial Model is proposed by Liu & Dietterich (2012) to depict the generation process, but the candidate labels are assumed to be instance-independent.

Deep PLL has recently been studied and advanced the practical application of PLL, where the PLL approaches are not restricted to linear models and low-efficiency optimization. Yao et al. (2020a) pioneer the use of deep convolutional neural networks and employ a regularization term of uncertainty and a temporal-ensembling term to train the deep model. Lv et al. (2020) propose a progressive identification method that allows PLL to be compatible with arbitrary models and optimizers while also performing impressively on image classification benchmarks for the first

time. Yao et al. (2020b) introduce a network-cooperation mechanism in deep PLL which trains two networks simultaneously to reduce their respective disambiguation errors. Feng et al. (2020) employ the importance reweighting strategy to derive a risk-consistent estimator and the transition matrix for PLL to derive a classifier-consistent estimator. Wen et al. (2021) introduce a leverage parameter that weights losses on partial labels and non-partial labels. Xu et al. (2021) apply variational label enhancement Xu et al. (2020; 2019a) to iteratively recover label distribution for each instance. Lv et al. (2021) for the first time propose unreliable PLL, a realistic and challenge setting. Zhang et al. (2021) discover class activation map could be used for disambiguation, and class activation value could be used to capture the learned representation information in a more general way. Lyu et al. (2022) transform the PLL problem into an "instance-label" matching selection problem. He et al. (2022) utilize semantic label representations to enhance the label disambiguation. Wang et al. (2022) create a contrastive learning framework for PLL disambiguation. To preserve the manifold structure in both feature and label space, Wu et al. (2022) employ manifold consistency regularization.

From Feng et al. (2020), the generation process of candidate labels has been paid attention to in Deep PLL, making a naive assumption that the candidate label set is uniformly sampled for each instance. Wen et al. (2021) extend the uniform case to the class-dependent one but remain instance-independent. Xu et al. (2021) for the first time consider PLL under the instance-dependent case, a more realistic but challenging setting. However, Xu et al. (2021) do not explicitly model the generation process of instance-dependent candidate labels. The generative model in it is employed only to generate label distribution. In this paper, we not only consider the instance-dependent PLL but also explicitly model the generation process of instance-dependent candidate labels via decoupled probability distributions.

## 3 PROPOSED METHOD

First of all, we introduce some necessary notations for our approach. In PLL, labeling information of every instance in the training dataset is corrupted from a correct label into a candidate label set, which contains the correct label and incorrect candidate labels but is not a complete label set. Let $\mathcal{X} \subseteq \mathbb{R}^q$ be the $q$-dimensional feature space of the instances and $\mathcal{Y} = \{1, 2, \ldots, c\}$ be the label space with $c$ class labels. Then a PLL training data set can be formulated as $\mathcal{D} = \{(\boldsymbol{x}_i, S_i) | 1 \le i \le n\}$, where $\boldsymbol{x}_i \in \mathcal{X}$ denotes the $i$-th $q$-dimensional feature vector and $S_i \in \mathcal{C} = 2^{\mathcal{Y}} \setminus \{\emptyset, \mathcal{Y}\}$ is the candidate label set annotated for $\boldsymbol{x}_i$. $S_i$ consists of the correct label $y_i \in \mathcal{Y}$ and the incorrect candidate label set $\overline{S}_i^{y_i} = S_i \setminus \{y_i\}$. We aim to find a multi-class classifier $f : \mathcal{X} \mapsto \mathcal{Y}$ according to $\mathcal{D}$.

### 3.1 OVERVIEW

To deal with instance-dependent PLL, we introduce the explicit generation model of instance-dependent candidate labels, which decouples the distribution of the correct and incorrect candidate labels. Then Category distribution and Bernoulli distribution are leveraged to depict them respectively with Dirichlet distribution and Beta distribution as their prior distributions. Based on the probabilistic generation model, we simply deduce the log-likelihood loss function for optimization and then move forward to the MAP optimization problem in consideration of the prior distributions Dirichlet and Beta. Finally, we further propose the algorithm IDGP, which keeps and refines the prior information.

### 3.2 GENERATION MODEL OF CANDIDATE LABELS

Focusing on instance-dependent partial label learning, we propose a novel generation model of candidate labels, which explicitly models the generation process and decouples the distribution of the candidate labels into the distribution of the correct label and incorrect candidate labels respectively. To form a candidate label set, we suppose the correct label is first selected according to a posterior distribution dependent on the instance. And then the incorrect candidate labels related to the feature of instance emerge to disturb the annotator and are sampled from another posterior distribution dependent on the instance. The concrete generation model is demonstrated as follows.

Given an instance $\boldsymbol{x}_i$, its candidate label set $S_i$, consisting of the correct label $y_i$ and the incorrect candidate labels $\overline{S}_i^{y_i}$, is drawn from a probability distribution with the following density:

$$p(S_i | \boldsymbol{x}_i) = \sum_{j \in S_i} p(y_i = j | \boldsymbol{x}_i) p(\overline{S}_i^j | \boldsymbol{x}_i). \tag{1}$$

Here, $p(S_i|\boldsymbol{x}_i)$ suggests our generation model is entirely dependent on the instance. Our generation model builds upon the PLL assumption that the correct label is always in the candidate set, which allows for directly splitting $S_i$ into $y_i$ and $\overline{S}_i^{y_i}$. For the given candidate label set $S_i$, each label $j \in S_i$ have the possibility $p(y_i = j|\boldsymbol{x}_i)$ sampled by the annotator as the correct label in the first stage of generation, and then the incorrect candidate labels $\overline{S}_i^j$ are sampled with the possibility $p(\overline{S}_i^j|\boldsymbol{x}_i)$ in the second stage to form the entire candidate label set $S_i$. The generation of the correct label and incorrect labels are conditioned on the instance $\boldsymbol{x}_i$. In this way, we decompose the generation process of the instance-dependent candidate labels.

After the decomposition of generation, the corresponding probability distributions are required to depict $p(y_i|\boldsymbol{x}_i)$ and $p(\overline{S}_i^{y_i}|\boldsymbol{x}_i)$. We assume that the correct label $y_i$ of each instance $\boldsymbol{x}_i$ is drawn from a Categorical distribution with the parameters $\boldsymbol{\theta}_i$, where $\boldsymbol{\theta}_i = [\theta_i^1, \theta_i^2, \ldots, \theta_i^c] \in [0,1]^c$ is a $c$-dimension vector with the constraint $\sum_{j=1}^c \theta_i^j = 1$, i.e.,

$$p(\boldsymbol{l}_i|\boldsymbol{x}_i, \boldsymbol{\theta}_i) = Cat(\boldsymbol{l}_i|\boldsymbol{\theta}_i) = \prod_{j=1}^c (\theta_i^j)^{l_i^j}, \tag{2}$$

where the vector $\boldsymbol{l}_i = [l_i^1, l_i^2 \ldots, l_i^c] \in \{0,1\}^c$ is utilized to represent whether the $j$-th label is the correct label $y_i$, i.e., $l_i^j = 1$ if $j = y_i$, otherwise $l_i^j = 0$.

Besides, for the incorrect candidate label set $\overline{S}_i^{y_i}$, it can also be denoted by a logical label vector $\overline{\boldsymbol{s}}_i^{y_i} = [\overline{s}_i^1, \overline{s}_i^2 \ldots, \overline{s}_i^c] \in \{0,1\}^c$ to represent whether the $j$-th label is the incorrect candidate label, i.e., $\overline{s}_i^j = 1$ if $j \in \overline{S}_i^{y_i}$, $\overline{s}_i^j = 0$ if $j \notin \overline{S}_i^{y_i}$. In order to describe $p(\overline{S}_i^{y_i}|\boldsymbol{x}_i)$ with a probabilistic model, we further decouple the incorrect candidate labels by assuming that the distribution of each variable $\overline{s}_i^j$ is independent from each other, i.e, $p(\overline{\boldsymbol{s}}_i^{y_i}|\boldsymbol{x}_i) = \prod_{j \in \overline{S}_i^{y_i}} p(\overline{s}_i^j|\boldsymbol{x}_i) \prod_{j \notin \overline{S}_i^{y_i}}(1 - p(\overline{s}_i^j|\boldsymbol{x}_i))$ and the incorrect candidate label set $\overline{S}_i^{y_i}$ is drawn from a multivariate Bernoulli distribution with the parameters $\boldsymbol{z}_i$, where $\boldsymbol{z}_i = [z_i^1, z_i^2, \ldots, z_i^c] \in [0,1]^c$ is a c-dimension vector, i.e.,

$$p(\overline{\boldsymbol{s}}_i^{y_i}|\boldsymbol{x}_i, \boldsymbol{z}_i) = \prod_{j=1}^c Ber(\overline{s}_i^j|z_i^j) = \prod_{j=1}^c (z_i^j)^{\overline{s}_i^j}(1 - z_i^j)^{1-\overline{s}_i^j}. \tag{3}$$

For the latter estimation of $\boldsymbol{\theta}_i$ in MAP, we introduce Dirichlet distribution, the conjugate distribution of the Categorical distribution, as its conditional prior, i.e., $p(\boldsymbol{\theta}_i|\boldsymbol{x}_i) = Dir(\boldsymbol{\theta}_i|\boldsymbol{\lambda}_i)$, where $\boldsymbol{\lambda}_i = [\lambda_i^1, \lambda_i^2, \ldots, \lambda_i^c]^\top (\lambda_i^j > 0)$ is a $c$-dimensional vector as the output of our main branch inference model $f$ parameterized by $\boldsymbol{\Theta}$ for an instance $\boldsymbol{x}_i$, i.e, $\boldsymbol{\lambda}_i = a \cdot \exp\left(f(\boldsymbol{x}_i; \boldsymbol{\Theta})/\gamma\right) + b$, where $a$, $b$ and $\gamma(a \geq 1, b \geq 0, \gamma > 0)$ are used to resolve the scale ambiguity. Note that the main branch inference model $f$ is leveraged as the final predictive model to accomplish the PLL target $\mathcal{X} \mapsto \mathcal{Y}$ by using $\widehat{y}_i = \arg\max_j \widehat{\theta}_i^j$, where $\widehat{\theta}_i^j$ will be estimated by $\boldsymbol{\lambda}_i$ with the conjugacy later.

Likewise, to estimate $\boldsymbol{z}_i$ latter in MAP, we introduce Beta distribution, the conjugate distribution of Bernoulli distribution, as the conditional prior parameterized by $\boldsymbol{\alpha}_i = [\alpha_i^1, \alpha_i^2, \ldots, \alpha_i^c]^\top$ and $\boldsymbol{\beta}_i = [\beta_i^1, \beta_i^2, \ldots, \beta_i^c]^\top (\alpha_i^j > 0, \beta_i^j > 0)$, i.e., $p(\boldsymbol{z}_i|\boldsymbol{x}_i) = \prod_{j=1}^c Beta(z_i^j|\alpha_i^j, \beta_i^j)$. We employ an auxiliary branch model $g$ parameterized by $\boldsymbol{\Omega}$ to output $\boldsymbol{\Lambda}_i^\top = [\boldsymbol{\alpha}_i^\top, \boldsymbol{\beta}_i^\top]$, i.e., $\boldsymbol{\Lambda}_i = a \cdot \exp\left(g(\boldsymbol{x}_i; \boldsymbol{\Omega})/\gamma\right) + b$. The same model and constants to scale $\boldsymbol{\Lambda}_i$ are implemented to simplify our approaches. It should be noted that the predictive model $f(\boldsymbol{x}_i; \boldsymbol{\Theta})$ or auxiliary model $g(\boldsymbol{x}_i; \boldsymbol{\Omega})$ can be any deep neural network once its output satisfies the corresponding constraint.

## 3.3 OPTIMIZATION USING MAXIMUM A POSTERIOR

Based on the generation model of candidate labels denoted by Eq.(1), Eq.(2) and Eq.(3), by using the technique of Maximum Likelihood (ML) estimation we can immediately induce the log-likelihood loss function for the PLL training data set to be optimized:

$$\mathcal{L}_{\mathrm{ML}} = -\sum_{i=1}^n \log p(S_i|\boldsymbol{x}_i, \boldsymbol{\theta}_i, \boldsymbol{z}_i) = -\sum_{i=1}^n \log \sum_{j \in S_i} \theta_i^j \prod_{k \in \overline{S}_i^j} z_i^k \prod_{k \notin \overline{S}_i^j}(1 - z_i^k). \tag{4}$$

The Eq.(4) demonstrates how the distribution of the correct label interacts with that of the incorrect candidate labels after decoupling the generation. In the backpropagation of the training process, they provide weight coefficients for each other.

In order to bring in prior information for PLL, we further introduce IDGP, which performs MAP in the training data set. In IDGP, we are more concerned about maximizing the joint distribution $p(\boldsymbol{\theta}, \boldsymbol{z}|\boldsymbol{x}, S)$. This will lead to the following optimization problem $\mathcal{L}_{\text{MAP}} = \mathcal{L}_{\text{ML}} + \mathcal{L}_{\text{reg}}$, where

$$\mathcal{L}_{\text{reg}} = -\sum_{i=1}^{n} \log p(\boldsymbol{\theta}_i|\boldsymbol{x}_i) + \log p(\boldsymbol{z}_i|\boldsymbol{x}_i). \tag{5}$$

Compared to ML, our IDGP framework provides a natural way of leveraging prior information via optimizing the extra condition prior illustrated by the Eq.(5) in the training process, which is significant for PLL due to the implicit supervision information. Combined with the prior distribution Dirichlet and Beta, $\mathcal{L}_{\text{reg}}$ can be analytically calculated as follows:

$$\mathcal{L}_{\text{reg}} = -\sum_{i=1}^{n}\sum_{j=1}^{c}(\lambda_i^j - 1)\log\theta_i^j + (\alpha_i^j - 1)\log z_i^j + (\beta_i^j - 1)\log(1 - z_i^j). \tag{6}$$

The mathematical derivations of Eq.(4) and Eq.(6) are provided in Appendix A.1. By combining the Eq.(4) and the Eq.(6), the MAP optimization problem $\mathcal{L}$ can be calculated as follows:

$$\mathcal{L} = -\sum_{i=1}^{n}\log\sum_{j\in S_i}\theta_i^j\prod_{k\in\overline{S}_i^j}z_i^k\prod_{k\notin\overline{S}_i^j}(1 - z_i^k)$$
$$-\sum_{i=1}^{n}\sum_{j=1}^{c}(\lambda_i^j - 1)\log\theta_i^j + (\alpha_i^j - 1)\log z_i^j + (\beta_i^j - 1)\log(1 - z_i^j). \tag{7}$$

$\mathcal{L}$ can also accommodate the uniform case, which can be seen in Appendix A.2. Then, due to the conjugacy of Dirichlet and Categorical distribution Minka (2000), we can estimate $\theta_i^j$ by using

$$\widehat{\theta}_i^j = \mathbb{E}\left[\theta_i^j|\boldsymbol{x}_i, \boldsymbol{\lambda}_i\right] = \frac{o_i^j + \lambda_i^j}{\sum_{k=1}^{c}\lambda_i^k + o_i^k}, \tag{8}$$

where $o_i^j$ denotes the number of occurrences on the label $j$ for $\boldsymbol{x}_i$, i.e., $o_i^j = 1$ if $j \in S_i$, otherwise $o_i^j = 0$. Similarly, we can leverage the conjugacy of Beta and Bernoulli distribution to estimate $z_i^j$, i.e,

$$\widehat{z}_i^j = \mathbb{E}\left[z_i^j|\boldsymbol{x}_i, \alpha_i^j, \beta_i^j\right] = \frac{o_i^j + \alpha_i^j}{\alpha_i^j + \beta_i^j + o_i^j}. \tag{9}$$

The mathematical derivations of Eq.(8) and Eq.(9) are provided in Appendix A.3.

As it is shown in the Eq.(6), IDGP provides prior information that can be included in the parameters $\lambda, \alpha$ and $\beta$ for the predictive model $f(\boldsymbol{x}; \boldsymbol{\Theta})$ and the auxiliary model $g(\boldsymbol{x}; \boldsymbol{\Omega})$ by transforming them into the weights exerted on the corresponding label. The prior information comes from the memorization effects Han et al. (2020) of the neural network. It makes the neural network always likely to recognize and remember the correct label in priority, leading to a kind of initial disambiguation at the beginning epochs. Hence, to keep fine prior information, we replace $\boldsymbol{\lambda}_i$ with $\widehat{\boldsymbol{\lambda}}_i$ in the following way:

$$\widehat{\lambda}_i^{j(t)} = \begin{cases} m\lambda_i^{j(r)} + (1-m)\lambda_i^{j(t)}, \text{if } j \in S_i \\ 1 + \epsilon, \text{otherwise}, \end{cases} \tag{10}$$

where $(\cdot)^{(t)}$ denotes the vector or scalar $(\cdot)$ at the $t$ epoch, $r(\leq t)$ denotes the beginning epoch we set to reserve the fine prior information for $f(\boldsymbol{x}; \boldsymbol{\Theta})$, $m \in (0, 1)$ is a positive constant used to replenish the present information $\lambda_i^{j(t)}$ with the prior information $\lambda_i^{j(r)}$, and $\epsilon$ is a minor value, which means that the weight exerted on each incorrect label is negligible. In the similar way, we replace $\boldsymbol{\alpha}_i^{(t)}$ and $\boldsymbol{\beta}_i^{(t)}$ with $\widehat{\boldsymbol{\alpha}}_i^{(t)}$ and $\widehat{\boldsymbol{\beta}}_i^{(t)}$, i.e, the vectors $\widehat{\boldsymbol{\alpha}}_i^{(t)}$ and $\widehat{\boldsymbol{\beta}}_i^{(t)}$ will be calculated by

$$\begin{cases} \widehat{\alpha}_i^{j(t)} = d\alpha_i^{j(q)} + (1-d)\alpha_i^{j(t)} \\ \widehat{\beta}_i^{j(t)} = d\beta_i^{j(q)} + (1-d)\beta_i^{j(t)}, \end{cases} \tag{11}$$

---

**Algorithm 1** IDGP Algorithm

---

**Input**: PLL training dataset $\mathcal{D} = \{(\boldsymbol{x}_i, S_i | 1 \le i \le n\}$, Epoch $\mathcal{T}$, Iteration $\mathcal{K}$;
**Output**: The predictive model $f(\boldsymbol{x}; \boldsymbol{\Theta})$
 1: Initialize the parameters of the predictive model $f(\boldsymbol{x}; \boldsymbol{\Theta})$ and the auxiliary model $g(\boldsymbol{x}; \boldsymbol{\Omega})$;
 2: Initialize $\widehat{\boldsymbol{\lambda}}_i^{(1)}, \widehat{\boldsymbol{\alpha}}_i^{(1)}, \widehat{\boldsymbol{\beta}}_i^{(1)}$ for each instance $\boldsymbol{x}_i$;
 3: **for** $t = 1, 2, \dots, \mathcal{T}$ **do**
 4:     Randomly shuffle the training dataset $\mathcal{D}$ and divide it into $\mathcal{K}$ mini-batches;
 5:     **for** $k = 1, 2, \dots, \mathcal{K}$ **do**
 6:         Calculate the parameters of Categorical distribution and Bernoulli distribution $\widehat{\boldsymbol{\theta}}_i$ and $\widehat{\boldsymbol{z}}_i$ for the instance $\boldsymbol{x}_i$ by the Eq.(8) and (9);
 7:         Calculate the parameters $\widehat{\boldsymbol{\lambda}}_i^{(t)}, \widehat{\boldsymbol{\alpha}}_i^{(t)}, \widehat{\boldsymbol{\beta}}_i^{(t)}$ for the instance $\boldsymbol{x}_i$ by the Eq.(10) and (11);
 8:         Fix $\boldsymbol{\Theta}$, update $\boldsymbol{\Omega}$ by the Eq.(7);
 9:         Fix $\boldsymbol{\Omega}$, update $\boldsymbol{\Theta}$ by the Eq.(7);
10:     **end for**
11: **end for**

---

where $q$ denotes the beginning epoch we set to reserve the fine prior information for the auxiliary model $g(\boldsymbol{x}; \boldsymbol{\Omega})$, and $d \in (0, 1)$ is also a positive constant used to provide the present model information $\alpha_i^{j(t)}, \beta_i^{j(t)}$ with the prior information $\alpha_i^{j(q)}, \beta_i^{j(q)}$. Before the epoch $r$, $\widehat{\lambda}_i^{j(t)} = \lambda_i^{j(t)}$ if $j \in S_i$, otherwise $1 + \epsilon$. Before the epoch $q$, $\widehat{\alpha}_i^{j(t)} = \alpha_i^{j(t)}$ and $\widehat{\beta}_i^{j(t)} = \beta_i^{j(t)}$. In the above way, we refine the prior information epoch by epoch.

The optimization of the two model $f(\boldsymbol{x}; \boldsymbol{\Theta})$ and $g(\boldsymbol{x}; \boldsymbol{\Omega})$ at the epoch $t$ for an instance $\boldsymbol{x}$ is as follows. First, $f(\boldsymbol{x}; \boldsymbol{\Theta})$ outputs $\boldsymbol{\lambda}$ while $g(\boldsymbol{x}; \boldsymbol{\Omega})$ outputs $\boldsymbol{\alpha}$ and $\boldsymbol{\beta}$, based on which, according to Eq.(8) and Eq.(9), we replace $\boldsymbol{\theta}$ and $\boldsymbol{z}$ with the estimation $\widehat{\boldsymbol{\theta}}$ and $\widehat{\boldsymbol{z}}$ in Eq.(7). Next, to introduce prior information, we use $\widehat{\boldsymbol{\lambda}}$, $\widehat{\boldsymbol{\alpha}}$ and $\widehat{\boldsymbol{\beta}}$ in Eq.(10) and Eq.(11) to replace $\boldsymbol{\lambda}$, $\boldsymbol{\alpha}$ and $\boldsymbol{\beta}$ in Eq.(7). Note that $\widehat{\boldsymbol{\theta}}$ and $\widehat{\boldsymbol{z}}$ are variables, of which we calculate the gradient to perform backpropagation, while $\widehat{\boldsymbol{\lambda}}$, $\widehat{\boldsymbol{\alpha}}$ and $\widehat{\boldsymbol{\beta}}$ are constants. Finally, we update the predictive model $f(\boldsymbol{x}; \boldsymbol{\Theta})$ and the auxiliary model $g(\boldsymbol{x}; \boldsymbol{\Omega})$ by fixing one and updating the other. The whole algorithmic description of the IDGP is shown in Algorithm 1. After implementing IDGP in the PLL training dataset, we can use the output $\boldsymbol{\lambda}$ of the main branch inference model $f(\boldsymbol{x}; \boldsymbol{\Theta})$ to calculate the predict results $\widehat{\boldsymbol{\theta}}$ as the label confidence in the test dataset.

## 4 THEORETICAL ANALYSIS

In this section, we pay attention to the estimation error bound of the predictive model $f(\boldsymbol{x}; \boldsymbol{\Theta})$. According to Eq.(7), the empirical risk estimator for the predictive model $f(\boldsymbol{x}; \boldsymbol{\Theta})$ is denoted by $\widehat{R}(f) = \frac{1}{n} \sum_{i=1}^{n} \mathcal{L}_{\text{MAP}}(f(\boldsymbol{x}_i), S_i)$ and for further analysis, we give an upper bound of $\mathcal{L}_{\text{MAP}}(f(\boldsymbol{x_i}), S_i)$ as follows:

$$\mathcal{L}_{\text{MAP}}(f(\boldsymbol{x}_i), S_i) \le - \left( K_i + \sum_{j=1}^{c} w_i^j \ell(f(\boldsymbol{x}_i), \boldsymbol{e}^j) \right), \tag{12}$$

where $\boldsymbol{e}^j$ denotes the standard canonical vector in $\mathbb{R}^c$, $\ell$ denotes the cross-entropy function, $w_i^j = \lambda_i^j - 1 + \frac{1}{|S_i|}$ if $j \in S_i$, otherwise $w_i^j = \lambda_i^j - 1$, and $K_i = \log |S_i| + \frac{1}{|S_i|} \sum_{j \in S_i} \log \prod_{k \in \overline{S}_i^j} z_i^k \prod_{k \notin \overline{S}_i^j} (1 - z_i^k) + (\alpha_i^j - 1) \log z_i^j + (\beta_i^j - 1) \log (1 - z_i^j)$. We denote $K = \max\{K_1, K_2, ..., K_n\}$ and scale $w_i^j$ to $[0, \rho]$ during the training process. The detailed induction of Eq.(12) can be seen in Appendix A.4. Then to formulate the estimation error bound of $f$, we give the following definition and lemmas.

**Definition 1.** *Let $\boldsymbol{x}_1, \boldsymbol{x}_2, \dots, \boldsymbol{x}_n \in \mathcal{X}$ be $n$ i.i.d random variables drawn from a probability distribution, $\sigma_1, \sigma_2, \dots, \sigma_n \in \{-1, +1\}$ be Rademacher variables with even probabilities, and $\mathcal{H} = \{h : \mathcal{X} \mapsto \mathbb{R}\}$ be a class of measurable functions. Then the expected Rademacher complexity of $\mathcal{H}$ is defined as*

$$\mathfrak{R}_n(\mathcal{H}) = \mathbb{E}_{\boldsymbol{x}, \sigma} \left[ \sup_{h \in \mathcal{H}} \frac{1}{n} \sum_{i=1}^{n} \sigma_i h(\boldsymbol{x}_i) \right].$$

Table 1: Classification accuracy (mean±std) of each comparing approach on benchmark datasets for instance-dependent PLL

| Datasets | MNIST | FMNIST | KMNIST | CIFAR10 | CIFAR100 |
|---|---|---|---|---|---|
| IDGP | **98.92 ± 0.05%** | **91.48 ± 0.32%** | **96.88 ± 0.17%** | **87.57 ± 0.26%** | **64.59 ± 0.17%** |
| PLCR | 98.56±0.08%• | 90.10±0.21%• | 95.29±0.21%• | 86.37±0.38%• | 64.12±0.23%• |
| PICO | 98.61±0.12%• | 88.41±0.20%• | 94.78±0.19%• | 86.16±0.21%• | 62.98±0.38%• |
| VALEN | 98.72±0.05%• | 90.63±0.30%• | 96.19±0.75% | 85.48±0.62%• | 62.96±0.96%• |
| CAVL | 98.84±0.05%• | 87.94±0.19%• | 93.69±0.28%• | 59.67±3.30%• | 52.59±1.01%• |
| LWS | 98.56±0.06%• | 88.99±0.26%• | 92.27±1.03%• | 37.49±2.82%• | 53.98±0.99%• |
| RC | 98.41±0.09%• | 89.60±0.19%• | 93.78±0.17%• | 85.95±0.40%• | 63.41±0.56%• |
| CC | 98.16±0.14%• | 89.86±0.11%• | 94.08±0.35%• | 79.96±0.99%• | 62.40±0.84%• |
| PRODEN | 98.39±0.10%• | 89.79±0.24%• | 93.79±0.24%• | 86.04±0.21%• | 62.56±1.49%• |

According Definition 1, given the function space $\mathcal{G} = \{(\boldsymbol{x}, S) \mapsto \mathcal{L}_{\text{MAP}}(f(\boldsymbol{x}), S) | f \in \mathcal{F}\}$, the expected Rademacher complexity of $\mathcal{G}$ can be defined as follows:

$$\widetilde{\mathfrak{R}}_n(\mathcal{G}) = \mathbb{E}_{\boldsymbol{x}, S, \sigma} \left[ \sup_{g \in \mathcal{G}} \frac{1}{n} \sum_{i=1}^{n} \sigma_i g(\boldsymbol{x}_i, S_i) \right]. \tag{13}$$

We pre-limit both the main model $f$ and the auxiliary model $g$ by clamping their output to $[-A, A]$, and the loss $\mathcal{L}_{\text{MAP}}$ can be bounded, though it would not have extended to infinity in practice.

**Lemma 1.** *Suppose the loss function $\mathcal{L}_{MAP}(f(\boldsymbol{x}), S)$ is bounded by $M$, i.e., $M = \sup_{\boldsymbol{x} \in \mathcal{X}, S \in \mathcal{C}, f \in \mathcal{F}} \mathcal{L}_{MAP}(f(\boldsymbol{x}), S)$, then for any $\xi > 0$, with probability at least $1 - \xi$,*

$$\sup_{f \in \mathcal{F}} \left| R(f) - \widehat{R}(f) \right| \leq 2\mathfrak{R}_n(\mathcal{G}) + \frac{M}{2} \sqrt{\frac{\log \frac{2}{\xi}}{2n}}.$$

**Lemma 2.** *Assume the loss function $\ell(f(\boldsymbol{x}), \boldsymbol{e}^\iota)$ is L-Lipschitz with respect to $f(\boldsymbol{x})$ $(0 < L < \infty)$ for all $\iota \in \mathcal{Y}$. Let $\mathcal{H}_\iota = \{h : \boldsymbol{x} \mapsto f_\iota(\boldsymbol{x}) | f \in \mathcal{F}\}$ and $\mathfrak{R}_n(\mathcal{H}_\iota) = \mathbb{E}_{\boldsymbol{x}, \sigma} \left[ \sup_{h \in \mathcal{H}_\iota} \frac{1}{n} \sum_{i=1}^{n} h(\boldsymbol{x}_i) \right]$, then the following inequality holds:*

$$\widetilde{\mathfrak{R}}_n(\mathcal{G}) \leq \sqrt{2} \rho c L \sum_{\iota \in \mathcal{Y}} \mathfrak{R}_n(\mathcal{H}_\iota) + K.$$

The proof of Lemma 1 and 2 is provided in Appendix A.5 and A.6. Based on Lemma 1 and 2, we induce an estimation error bound for our IDGP method. Let $\widehat{f} = \arg\min_{f \in \mathcal{F}} \widehat{R}(f)$ be the empirical risk minimizer and $f^\star = \arg\min_{f \in \mathcal{F}} R(f)$ be the true minimizer. The function space $\mathcal{H}_\iota$ for the label $\iota \in \mathcal{Y}$ is defined as $\{h : x \mapsto f_y(x) | f \in \mathcal{F}\}$, and $\mathfrak{R}_n(\mathcal{H}_\iota)$ is defined as the expected Rademacher complexity of $\mathcal{H}_\iota$ with sample size $n$. Then we have the following theorem.

**Theorem 1.** *Assume the loss function $\ell(f(\boldsymbol{x}), \boldsymbol{e}^\iota)$ is L-Lipschitz with respect to $f(\boldsymbol{x})$ $(0 < L < \infty)$ for all $\iota \in \mathcal{Y}$ and $\mathcal{L}_{MAP}(f(\boldsymbol{x}), S)$ is bounded by $M$, i.e., $M = \sup_{\boldsymbol{x} \in \mathcal{X}, S \in \mathcal{C}, f \in \mathcal{F}} \mathcal{L}_{MAP}(f(\boldsymbol{x}), S)$,. Then, for any $\xi > 0$, with probability at least $1 - \xi$,*

$$R(\widehat{f}) - R(f^\star) \leq 4\sqrt{2} \rho c L \sum_{\iota \in \mathcal{Y}} \mathfrak{R}_n(\mathcal{H}_\iota) + M \sqrt{\frac{\log \frac{2}{\xi}}{2n}} + 4K.$$

The proof can be found in Appendix A.7. Theorem 1 means that as $n \to \infty$ the empirical risk minimizer $f$ will converge to the optimal risk minimizer $f^\star$ for models with a bounded norm.

## 5 EXPERIMENTS

In this section, we validate the effectiveness of our proposed IDGP by performing it on augmented benchmark and real-world datasets and comparing its results against DNN-based PLL algorithms. Furthermore, the ablation study and sensitive analysis of parameters are conducted to explore IDGP.

Table 2: Classification accuracy (mean±std) of comparing algorithms on the real-world datasets.

|  | Lost | BirdSong | MSRCv2 | Soccer Player | Yahoo!News |
|---|---|---|---|---|---|
| IDGP | **77.02±0.82%** | **74.23±0.17%** | **50.45±0.47%** | 55.99±0.28% | **66.62±0.19%** |
| VALEN | 76.87±0.86% | 73.39±0.26%● | 49.97±0.43%● | 55.81±0.10% | 66.26±0.13%● |
| CAVL | 75.89±0.42%● | 73.47±0.13%● | 44.73±0.96%● | 54.06±0.67%● | 65.44±0.23%● |
| LWS | 73.13±0.32%● | 51.45±0.26%● | 49.85±0.49%● | 50.24±0.45%● | 48.21±0.29%● |
| RC | 76.26±0.46% | 69.33±0.32%● | 49.47±0.43%● | **56.02±0.59%** | 63.51±0.20%● |
| CC | 63.54±0.25%● | 69.90±0.58%● | 41.50±0.44%● | 49.07±0.36%● | 54.86±0.48%● |
| PRODEN | 76.47±0.25% | 73.44±0.12%● | 45.10±0.16%● | 54.05±0.15%● | 66.14±0.10%● |

## 5.1 DATASETS

We implement IDGP with compared DNN-based algorithms on five widely used benchmark datasets in deep learning, including `MNIST` LeCun et al. (1998), `Kuzushiji-MNIST` Clanuwat et al. (2018), `Fashion-MINIST` Xiao et al. (2017), `CIFAR-10` and `CIFAR-100` Krizhevsky et al. (2009). Instance-dependent partial labels for these datasets are generated through the same strategy as Xu et al. (2021), which for the first time consider instance-dependent PLL.

Besides, part of the comparing algorithms are also performed on five frequently used real-world datasets, which come from different practical application domains, including `Lost` Cour et al. (2011), `BirdSong` Briggs et al. (2012), `MSRCv2` Liu & Dietterich (2012), `Soccer Player` Zeng et al. (2013) and `Yahoo!News` Guillaumin et al. (2010).

For benchmark datasets, we split $10\%$ samples from the training datasets for validating. For each real-world dataset, we run the methods with $80\%/10\%/10\%$ train/validation/test split. Then we run five trials on each datasets with different random seeds and report the mean accuracy and standard deviation of all comparing algorithms.

## 5.2 BASELINES

We compare IDGP with eight DNN-based methods:1) PLCR Wu et al. (2022), a regularized training framework which is based on data augmentation and utilizes the manifold consistency regularization term to preserve the manifold structure both in feature space and label space. 2) PICO Wang et al. (2022), a contrastive learning framework which is based on data augmentation and performs label disambiguation based on the contrastive prototypes. 3) VALEN Xu et al. (2021), an instance-dependent PLL framework which guides the training process via the recovered latent label distributions. 4) CAVL Zhang et al. (2021), a discriminative approach which identifies correct labels from candidate labels by class activation value. 5) LWS Wen et al. (2021), an identification-based method which introduces a leverage parameter to consider the trade-off between losses on candidate and non-candidate labels. 6) RC Feng et al. (2020), a risk-consistent PLL method which is induced by an importance reweighting strategy. 7) CC Feng et al. (2020), a classifier-consistent PLL method which leverages the transition matrix describing the probability of the candidate label set given a correct label. 8) PRODEN Lv et al. (2020), a self-training style algorithm which provides a framework to equip arbitrary stochastic optimizers and models in PLL. Note that PLCR and PICO will not be compared on the real-world datasets due to the requirement of data augmentation.

To ensure the fairness, we employ the same network backbone, optimizer and data augmentation strategy for all the comparing methods. For `MNIST`, `Kuzushiji-MNIST` and `Fashion-MNIST`, we take LeNet-5 as their backbone. For `CIFAR-10` and `CIFAR-100`, the network backbone is changed to ResNet-32 He et al. (2016). For all the real-world datasets, we simply adopt the linear model. The optimizer is stochastic gradient descent (SGD) Robbins & Monro (1951) with momentum $0.9$ and batch size $256$. The details of data augmentation strategy are shown in Appendix A.8. Besides, the learning rate is selected from $\{10^{-4}, 10^{-3}, 10^{-2}\}$, and the weight decay is selected from $\{10^{-5}, 10^{-4}, 10^{-3}, 10^{-2}\}$ according to the performance on the validation.

## 5.3 EXPERIMENTAL RESULTS

The performance of each DNN-based method on each corrupted benchmark dataset is summarized in Table 1, where the best results are highlighted in bold and ●/○ indicates whether IDGP statistically

Table 3: Classification accuracy (mean±std) for comparison against IDGP-ML.

| Dataset | IDGP | IDGP-ML |
|---|---|---|
| Lost | **77.02±0.82%** | 52.50±3.20%● |
| BirdSong | **74.23±0.17%** | 71.13±0.45%● |
| MSRCv2 | **50.45±0.47%** | 43.04±0.98%● |
| Soccer Player | **55.99±0.28%** | 49.56±0.31%● |
| Yahoo!News | **66.62±0.19%** | 51.48±0.07%● |

(a) Lost     (b) BirdSong     (c) Yahoo!News

Figure 1: Parameter sensitivity analysis for IDGP on Lost, BirdSong and Yahoo!New

wins/loses to the comparing method on each dataset additionally (pairwise t-test at 0.05 significance level). We can overall see that IDGP significantly outperforms all comparing approaches on all benchmark datasets (except on Kuzushiji-MNIST where VALEN performs comparably against IDGP), and the improvements are particularly noticeable on Fashion-MNIST and CIFAR-10.

Table 2 demonstrates the ability of IDGP to solve the PLL problem in real-world datasets. PLCR and PICO are not compared on the real-world PLL datasets due to the inability of data augmentation to be employed on the extracted features from various domains. We can find that our method has stronger competence than others in all datasets except Soccer Player where IDGP loses to RC but still ranks second. As for BirdSong, MSRCv2 and Yahoo!News, the performance of IDGP is significantly better than all other comparing algorithms. And when it comes to Lost, our method is comparable to VALEN, PRODEN and RC, while obviously better than the rest.

## 5.4 FURTHER ANALYSIS

To demonstrate the effectiveness of the iteratively refined prior information introduced by IDGP, we remove the loss function $\mathcal{L}_{reg}$ to reverse IDGP to IDGP-ML, which only uses the log-likelihood function for optimization. The performance of IDGP-ML against IDGP is also measured by the classification accuracy (with pairwise t-test at 0.05 significance level). As is illustrated in Table 3, with the assistance of the prior information which IDGP provides and improves epoch by epoch, IDGP achieves superior performance on all real-world datasets compared to IDGP-ML.

Furthermore, we conduct parameter sensitivity analysis to study the influence of the two hyper-parameters $a, \gamma$ on our algorithm, which decides the scale of Dirichlet and Beta distribution parameters. Figure 2 illustrates the sensitivity of IDGP in the real-world datasets including Lost, BirdSong, Yahoo!New when $a$ varies from 0.001 to 1000 and $\gamma$ increases from 0.1 to 3. We can easily find that as for the small-scale real-world datasets like Lost, $a$ and $\gamma$ are suggested around 0.1 and 0.5, respectively. For the more large-scale real-world datasets like BirdSong, Yahoo!News, IDGP seems insensitive to $a$, and $\gamma$ are suggested around 1.

## 6 CONCLUSION

In this paper, we consider a more realistic scenario, instance-dependent PLL, and explicitly decompose and model the generation process of instance-dependent candidate labels. Then based on the decompositional generation process, a novel instance-dependent PLL approach IDGP is proposed by us to further introduce and refine the prior information in every training epoch via MAP. The experimental comparisons with other DNN-based algorithms on both instance-dependent corrupted benchmark datasets and real-world datasets demonstrate the effectiveness of our proposed method.

ACKNOWLEDGMENTS

This research was supported by the National Key Research & Development Plan of China (No. 2021ZD0114202), the National Science Foundation of China (62206050, 62125602, and 62076063), China Postdoctoral Science Foundation (2021M700023), Jiangsu Province Science Foundation for Youths (BK20210220), Young Elite Scientists Sponsorship Program of Jiangsu Association for Science and Technology (TJ-2022-078).

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

## A  APPENDIX

### A.1  DERIVATIONS OF EQ.(4) AND EQ.(6)

The derivation of Eq.(4) is as follows:

$$
\begin{aligned}
\mathcal{L}_{\mathrm{ML}} &= -\sum_{i=1}^{n} \log p(S_i | \boldsymbol{x}_i, \boldsymbol{\theta}_i, \boldsymbol{z}_i) \\
&= -\sum_{i=1}^{n} \log \sum_{j \in S_i} p(y_i = j | \boldsymbol{x}_i) p(\bar{S}_i^j | \boldsymbol{x}_i) \\
&= -\sum_{i=1}^{n} \log \sum_{j \in S_i} \left( Cat(\boldsymbol{l}_i | \boldsymbol{x}_i, \boldsymbol{\theta}_i) \prod_{k=1}^{c} Ber(s_i^{\bar{k}} | z_i^k) | y_i = j \right) \\
&= -\sum_{i=1}^{n} \log \sum_{j \in S_i} \left( \prod_{k=1}^{c} (\theta_i^k)^{l_i^k} \prod_{k=1}^{c} (z_i^k)^{s_i^{\bar{k}}} (1 - z_i^k)^{1 - s_i^{\bar{k}}} | y_i = j \right) \\
&= -\sum_{i=1}^{n} \log \sum_{j \in S_i} \theta_i^j \prod_{k \in \bar{S}_i^j} z_i^k \prod_{k \notin \bar{S}_i^j} (1 - z_i^k).
\end{aligned}
\tag{14}
$$

The derivation of Eq.(6) is as follows:

$$
\begin{aligned}
\mathcal{L}_{\mathrm{reg}} &= -\sum_{i=1}^{n} \log p(\boldsymbol{\theta}_i | \boldsymbol{x}_i) + \log p(\boldsymbol{z}_i | \boldsymbol{x}_i) \\
&= -\sum_{i=1}^{n} \log \frac{\Gamma(\sum_{j=1}^{c} \lambda_i^j)}{\prod_{j=1}^{c} \Gamma(\lambda_i^j)} \prod_{j=1}^{c} (\theta_i^j)^{\lambda_i^j - 1} + \log \prod_{j=1}^{c} \frac{\Gamma(\alpha_i^j + \beta_i^j)}{\Gamma(\alpha_i^j)\Gamma(\beta_i^j)} (z_i^j)^{\alpha_i^j - 1} (1 - z_i^j)^{\beta_i^j - 1} \\
&= -\sum_{i=1}^{n} \log \prod_{j=1}^{c} (\theta_i^j)^{\lambda_i^j - 1} + \log \prod_{j=1}^{c} (z_i^j)^{\alpha_i^j - 1} (1 - z_i^j)^{\beta_i^j - 1} + C_\Gamma \\
&= -\sum_{i=1}^{n} \sum_{j=1}^{c} (\lambda_i^j - 1) \log(\theta_i^j) + (\alpha_i^j - 1) \log(z_i^j) + (\beta_i^j - 1) \log(1 - z_i^j) + C_\Gamma.
\end{aligned}
\tag{15}
$$

$C_\Gamma = -\sum_{i=1}^{n} \log \frac{\Gamma(\sum_{j=1}^{c} \lambda_i^j)}{\prod_{j=1}^{c} \Gamma(\lambda_i^j)} + \log \prod_{j=1}^{c} \frac{\Gamma(\alpha_i^j + \beta_i^j)}{\Gamma(\alpha_i^j)\Gamma(\beta_i^j)}$. Due to that we later use $\hat{\boldsymbol{\lambda}}$, $\hat{\boldsymbol{\alpha}}$ and $\hat{\boldsymbol{\beta}}$ in Eq.(10) and Eq.(11) to replace $\boldsymbol{\lambda}$, $\boldsymbol{\alpha}$ and $\boldsymbol{\beta}$ in Eq.(7), which are fixed to be constants, $C_\Gamma$ will be a constant and ignored. Here, the derivation of Eq.(6) has been finished.

## A.2 Degeneration of Eq.(7)

The proposed process can accommodate the uniform generation process of candidate labels by settng the parameters of Bernoulli distribution to a constant $p$, which also means the flipping probability. In this case, we can degenerate $\mathcal{L}_{\text{ML}}, \mathcal{L}_{\text{reg}}$ and $\mathcal{L}$ to the following form.

$$
\begin{aligned}
\mathcal{L}_{\text{ML-D}} &= -\sum_{i=1}^{n} \log \sum_{j \in S_i} \theta_i^j \prod_{k \in \bar{S}_i^j} z_i^k \prod_{k \notin \bar{S}_i^j} (1 - z_i^k) \\
&= -\sum_{i=1}^{n} \log \sum_{j \in S_i} \theta_i^j (1-p)^{c+1-|S_i|} p^{|S|-1} \\
&= -\sum_{i=1}^{n} \log (1-p)^{c+1-|S_i|} p^{|S|-1} \sum_{j \in S_i} \theta_i^j \\
&= -\sum_{i=1}^{n} \log \sum_{j \in S_i} \theta_i^j - \sum_{i=1}^{n} \log (1-p)^{c+1-|S_i|} p^{|S|-1}.
\end{aligned}
\tag{16}
$$

Due to that the second term is a constant, the final $\mathcal{L}_{\text{ML}}$ is formulated as

$$
\mathcal{L}_{\text{ML-D}} = -\sum_{i=1}^{n} \log \sum_{j \in S_i} \theta_i^j.
\tag{17}
$$

Due to the parameters of Bernoulli Distribution are constant, we do not need Beta Distribution. Hence,

$$
\mathcal{L}_{\text{reg-D}} = -\sum_{i=1}^{n} \sum_{j=1}^{c} (\lambda_i^j - 1) \log \theta_j^i.
\tag{18}
$$

Finally,

$$
\mathcal{L}_{\text{D}} = \mathcal{L}_{\text{ML-D}} + \mathcal{L}_{\text{reg-D}} = -\sum_{i=1}^{n} \log \sum_{j \in S_i} \theta_i^j - \sum_{i=1}^{n} \sum_{j=1}^{c} (\lambda_i^j - 1) \log \theta_j^i.
\tag{19}
$$

## A.3 Derivations of Eq.(8) and Eq.(9)

$$
\begin{aligned}
\boldsymbol{\theta}_i \sim p(\boldsymbol{\theta}_i | \boldsymbol{x}_i, \boldsymbol{\lambda}_i) &\propto Dir(\boldsymbol{\theta}_i | \boldsymbol{\lambda}_i) \cdot Cat(\boldsymbol{l}_i | \boldsymbol{\theta}_i) \\
&= \frac{\Gamma(\sum_{j=1}^{c} \lambda_i^j)}{\prod_{j=1}^{c} \Gamma(\lambda_i^j)} \cdot \prod_{j=1}^{c} (\theta_i^j)^{\lambda_i^j - 1} \cdot \prod_{j=1}^{c} (\theta_i^j)^{l_i^j} \\
&= \frac{\Gamma(\sum_{j=1}^{c} \lambda_i^j)}{\prod_{j=1}^{c} \Gamma(\lambda_i^j)} \cdot \prod_{j=1}^{c} (\theta_i^j)^{\lambda_i^j + l_i^j - 1} \\
&= Dir(\boldsymbol{\theta}_i | \boldsymbol{\lambda}_i + \boldsymbol{l}_i).
\end{aligned}
\tag{20}
$$

The above has proved the conjugacy of Dirichlet and Categorical distribution. For Dirichlet distribution, its expectation can be calculated as

$$
\mathbb{E}[\theta_i^j | \boldsymbol{x}_i, \boldsymbol{\lambda}_i] = \frac{\lambda_i^j + l_i^j}{\sum_{k=1}^{c} \lambda_i^k + l_i^k} = \frac{\lambda_i^j + o_i^j}{\sum_{k=1}^{c} \lambda_i^k + o_i^k}.
\tag{21}
$$

The mathematical derivations of Eq.(8) are completed. Eq.(9) can be proved in a similar way due to Dirichlet distribution and Categorical distribution are the generalization forms of Beta distribution and Bernoulli distribution respectively.

A.4 CALCULATION DETAILS OF EQ. (12)

According to multivariate basic inequality, we can obtain:

$$-\log \sum_{j \in S_i} \theta_i^j \prod_{k \in \overline{S}_i^j} z_i^k \prod_{k \notin \overline{S}_i^j} (1 - z_i^k) \leq -\log |S_i| - \frac{1}{|S_i|} \sum_{j \in S_i} \left( \log \theta_i^j + \log \prod_{k \in \overline{S}_i^j} z_i^k \prod_{k \notin \overline{S}_i^j} (1 - z_i^k) \right).$$

(22)

Then we can calculate Eq. (12) as

$$\mathcal{L}_{\text{MAP}} (f(\boldsymbol{x}_i), S_i) = -\log \sum_{j \in S_i} \theta_i^j \prod_{k \in \overline{S}_i^j} z_i^k \prod_{k \notin \overline{S}_i^j} (1 - z_i^k)$$

$$- \sum_{j=1}^{c} (\lambda_i^j - 1) \log \theta_i^j + (\alpha_i^j - 1) \log z_i^j + (\beta_i^j - 1) \log (1 - z_i^j)$$

$$\leq -\log |S_i| - \frac{1}{|S_i|} \sum_{j \in S_i} \left( \log \theta_i^j + \log \prod_{k \in \overline{S}_i^j} z_i^k \prod_{k \notin \overline{S}_i^j} (1 - z_i^k) \right)$$

$$- \sum_{j=1}^{c} (\lambda_i^j - 1) \log \theta_i^j + (\alpha_i^j - 1) \log z_i^j + (\beta_i^j - 1) \log (1 - z_i^j)$$

$$= - \left( \log |S_i| + \frac{1}{|S_i|} \sum_{j \in S_i} \log \prod_{k \in \overline{S}_i^j} z_i^k \prod_{k \notin \overline{S}_i^j} (1 - z_i^k) + \sum_{j=1}^{c} (\alpha_i^j - 1) \log z_i^j + (\beta_i^j - 1) \log (1 - z_i^j) \right)$$

$$- \sum_{j \in S_i} \left( \lambda_i^j - 1 + \frac{1}{|S_i|} \right) \log \theta_i^j - \sum_{j \notin S_i} \left( \lambda_i^j - 1 \right) \log \theta_i^j$$

$$= - \left( K_i + \sum_{j=1}^{c} w_i^j \ell(f(\boldsymbol{x}_i), \boldsymbol{e}^j) \right)$$

(23)

where $\boldsymbol{e}^j$ denotes the standard canonical vector in $\mathbb{R}^c$, and $\ell$ denotes the cross-entropy function, $w_i^j = \lambda_i^j - 1 + \frac{1}{|S_i|}$ if $j \in S_i$, otherwise $w_i^j = \lambda_i^j - 1$, and $K_i = \log |S_i| + \frac{1}{|S_i|} \sum_{j \in S_i} \log \prod_{k \in \overline{S}_i^j} z_i^k \prod_{k \notin \overline{S}_i^j} (1 - z_i^k) + (\alpha_i^j - 1) \log z_i^j + (\beta_i^j - 1) \log (1 - z_i^j)$.

A.5 PROOF OF LEMMA 1

In order to prove this lemma, we first show $\mathcal{L}_{\text{MAP}}$ can be bounded. Due to the ouput of $f, g$ is limited in $[-A, A]$, the following inequations hold:

$$\widehat{\theta}_i^j \geq \frac{a \exp(-A/\gamma) + b}{ac \exp(A/\gamma) + bc + c} = B$$

(24)

$$E = \frac{a \exp(-A/\gamma) + b}{2(a \exp(A/\gamma) + b) + 1} \leq \widehat{z}_i^j \leq \frac{a \exp(A/\gamma) + b + 1}{a \exp(A/\gamma) + a \exp(-A/\gamma) + 2b + 1} = F$$

(25)

Hence,

$$\mathcal{L}_{\text{MAP}} \leq -\log |S_i| B (1 - F)^{c+1-|S_i|} E^{|S_i|-1} - c \log [BE(1 - F)]^{a \exp(A/\gamma) + b}$$

When $M$ takes the value larger than $-\log |S_i| B (1 - F)^{c+1-|S_i|} E^{|S_i|-1} - c \log [BE(1 - F)]^{a \exp(A/\gamma) + b}$, the loss $\mathcal{L}_{\text{MAP}}$ will be bounded. Note that the limitation to the output of the model excludes extreme (conditional) probabilities, the effect of which could be ignored which $A$ is large enough.

Then, we show that the one direction $\sup_{f \in \mathcal{F}} R(f) - \widehat{R}(f)$ is bounded with probability at least $1 - \xi/2$, and the other direction can be similarly shown. Suppose an example $(\boldsymbol{x}_i, S_i)$ is replaced by another arbitrary example $(\boldsymbol{x}_i', S_i')$, then the change of $\sup_{f \in \mathcal{F}} R(f) - \widehat{R}(f)$ is no greater than $M/(2n)$, the loss function $\mathcal{L}$ are bounded by $M$. By applying McDiarmid's inequality, for any $\xi > 0$, with probability at least $1 - \xi/2$,

$$\sup_{f \in \mathcal{F}} R(f) - \widehat{R}(f) \le \mathbb{E}\left[\sup_{f \in \mathcal{F}} R(f) - \widehat{R}(f)\right] + \frac{M}{2}\sqrt{\frac{\log \frac{2}{\delta}}{2n}}. \tag{26}$$

By sysmetrization, we can obtain

$$\mathbb{E}\left[\sup_{f \in \mathcal{F}} R(f) - \widehat{R}(f)\right] \le 2\widetilde{\mathfrak{R}}_n(\mathcal{G}). \tag{27}$$

By further taking into account the other side $\sup_{f \in \mathcal{F}} R(f) - \widehat{R}(f)$, we have for any $\xi > 0$, with probability at least $1 - \xi$,

$$\sup_{f \in \mathcal{F}}|R(f) - \widehat{R}(f)| \le 2\mathfrak{R}_n(\mathcal{G}) + \frac{M}{2}\sqrt{\frac{\log \frac{2}{\xi}}{2n}} \tag{28}$$

### A.6 PROOF OF LEMMA 2

The upper bound loss function of $\mathcal{L}_{\text{MAP}}(f(\boldsymbol{x_i}), S_i)$ is denoted by

$$\bar{\mathcal{L}}(f(\boldsymbol{x_i}), S_i) = -\left(K_i + \sum_{j=1}^{c} w_i^j \ell(f(\boldsymbol{x}_i), \boldsymbol{e}^j)\right) \tag{29}$$

Correspondingly, the function space for $\bar{\mathcal{L}}$ can be defined as:

$$\bar{\mathcal{G}} = \left\{(\boldsymbol{x}, S) \mapsto \bar{\mathcal{L}}\left(f(\boldsymbol{x}), S\right) | f \in \mathcal{F}\right\} \tag{30}$$

Then the expected Rademacher complexity of $\bar{\mathcal{G}}$ can be defined as follows:

$$\widetilde{\mathfrak{R}}_n(\bar{\mathcal{G}}) = \mathbb{E}_{\boldsymbol{x}, S, \sigma}\left[\sup_{\bar{g} \in \bar{\mathcal{G}}} \frac{1}{n}\sum_{i=1}^{n} \sigma_i \bar{g}(\boldsymbol{x}_i, S_i)\right] \tag{31}$$

For each example $(\boldsymbol{x}_i, S_i)$, since $w_i^j$ is bounded in $[0, \rho]$ and $K_i \le K$, we can obtain $\widetilde{\mathfrak{R}}_n(\mathcal{G}) \le \widetilde{\mathfrak{R}}_n(\bar{\mathcal{G}}) \le \rho c \mathfrak{R}_n(\ell \circ \mathcal{F}) + K$ where $\ell \circ \mathcal{F}$ denotes $\{\ell \circ \mathcal{F} | f \in \mathcal{F}\}$. Since $\mathcal{H}_\iota = \{h : \boldsymbol{x} \mapsto f_y(\boldsymbol{x}) | f \in \mathcal{F}\}$ and the loss function $\ell\left(f(\boldsymbol{x}, \boldsymbol{e}^\iota)\right)$ is $L$-Lipschitz for all $\iota \in \mathcal{Y}$, by the Rademacher vector contraction inequality, we have $\mathfrak{R}_n(\ell \circ \mathcal{F}) \le \sqrt{2}L \sum_{\iota \in \mathcal{Y}} \mathfrak{R}_n(\mathcal{H}_\iota)$. Then the proof is completed.

### A.7 PROOF OF THEOREM 1

Theorem is proven through

$$\begin{aligned}
R(\widehat{f}) - R(f^*) &= R(\widehat{f}) - \widehat{R}(\widehat{f}) + \widehat{R}(\widehat{f}) - \widehat{R}(f^*) + \widehat{R}(f^*) - R(f^*) \\
&\le R(\widehat{f}) - \widehat{R}(\widehat{f}) + \widehat{R}(f^*) - R(f^*) \\
&\le 2\sup_{f \in \mathcal{F}}\left|R(f) - \widehat{R}(f)\right| \\
&\le 4\widetilde{\mathfrak{R}}_n(\mathcal{G}) + M\sqrt{\frac{\log \frac{2}{\delta}}{2n}} \\
&\le 4\sqrt{2}\rho c L \sum_{\iota \in \mathcal{Y}} \mathfrak{R}_n(\mathcal{H}_\iota) + M\sqrt{\frac{\log \frac{2}{\delta}}{2n}} + 4K
\end{aligned} \tag{32}$$

Table 4: Classification accuracy (mean±std) of each comparing approach on benchmark datasets, of which instance-dependent partial labels are generated by our generation model.

| Datasets | MNIST | FMNIST | KMNIST | CIFAR10 | CIFAR100 |
|---|---|---|---|---|---|
| IDGP | **98.87 ± 0.05**% | **91.20 ± 0.21**% | **97.28 ± 0.19**% | **90.18 ± 0.32**% | **64.67 ± 0.23**% |
| PLCR | 98.46±0.22%● | 90.18±0.27%● | 95.19±0.13%● | 87.07±0.34%● | 64.43±0.44% |
| PICO | 98.61±0.12%● | 88.32±0.17%● | 94.78±0.19%● | 86.56±2.74%● | 63.05±0.26%● |
| VALEN | 98.77±0.04%● | 90.58±0.36%● | 96.41±0.33%● | 86.31±0.37%● | 63.77±0.51%● |
| CAVL | 98.77±0.09%● | 88.09±0.32%● | 93.30±0.55%● | 60.20±2.64%● | 52.85±2.21%● |
| LWS | 98.52±0.14%● | 88.90±0.25%● | 92.69±0.76%● | 39.75±2.22%● | 53.52±1.37%● |
| RC | 98.35±0.14%● | 89.78±0.29%● | 94.06±0.05%● | 85.48±0.27%● | 63.24±0.97%● |
| CC | 98.15±0.07%● | 89.67±0.31%● | 94.12±0.17%● | 80.19±0.76%● | 62.11±1.03%● |
| PRODEN | 98.43±0.10%● | 89.72±0.36%● | 94.08±0.45%● | 85.60±0.20%● | 62.87±0.45%● |

Table 5: Classification accuracy (mean±std) for comparison against IDGP-ML on benchmark datasets, of which instance-dependent partial labels are generated by our generation model.

| Dataset | IDGP | IDGP-ML |
|---|---|---|
| MNIST | **98.87±0.05**% | 91.79±0.12%● |
| Kuzushiji-MNIST | **97.28±0.19**% | 76.67±0.34%● |
| Fashion-MINIST | **91.27±0.21**% | 78.38±0.29%● |
| CIFAR-10 | **90.18±0.32**% | 69.47±0.45%● |
| CIFAR-100 | **64.67±0.23**% | 40.23±0.57%● |

## A.8 DETAILS OF DATA AUGMENTATION

For all benchmark datasets, we apply Random Horizontal Flipping, Random Cropping, and Cutout. For `CIFAR-10` and `CIFAR-100`, AutoAugment is additionally applied.

## A.9 EXTENDING EXPERIMENTS

**Our Generation Model**. We also synthesize the sampled correct label from Categorical Distribution $p(y = j|\boldsymbol{x})$ and incorrect candidate labels from Bernoulli Distribution $p(\bar{s}^j = 1|\boldsymbol{x})$. For the former, correct labels has been contained in these datasets. For the latter, we use the confidence prediction of a clean neural network $g(\boldsymbol{x}; \hat{\boldsymbol{\Omega}})$ (trained only with correct labels) to model the Bernoulli Distribution, i.e., $p(\bar{s}^j = 1|\boldsymbol{x}) = \text{sigmoid}(g_j(\boldsymbol{x}; \hat{\boldsymbol{\Omega}}))$. Table 4 illustrates the performance of IDGP and comparing approaches on on benchmark datasets, of which instance-dependent partial labels generated by our generation model. From the table, we can overall see that IDGP consistently outperforms all comparing approaches on all benchmark datasets, and the improvements are particularly noticeable on `Kuzushiji-MNIST` and `CIFAR-10`.

**Ablation Study**. As is illustrated in Table 5, IDGP also achieves superior performance on all benchmark datasets, of which instance-dependent partial labels are generated by our generation model.

**Sensitive Analysis**. For the benchmark dataset `Fashion-MINIST`, the performance of IDGP is stable and effective in the case that $a$ is around 10 and $\gamma$ is around 1.5.

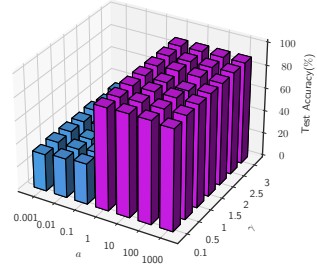

(a) `Fashion-MINIST`

Figure 2: Parameter sensitivity analysis for IDGP on `Fashion-MINIST`.

