# OpenReview forum: "Decompositional Generation Process for Instance-Dependent Partial Label Learning"
_ICLR.cc/2023/Conference — ICLR 2023 notable top 25%_

### Official Review · Reviewer_bU6i · 2022-10-22

**Confidence:** 4
**Correctness:** 3
**Technical Novelty And Significance:** 3
**Empirical Novelty And Significance:** 2
**Recommendation:** 6

**Clarity, Quality, Novelty And Reproducibility:**

Clarity issues discussed above.

Originality ... well, let's be honest, it's a obvious incremental next step given VALEN, but it's still cool.

Quality issues discussed above.

**Strength And Weaknesses:**

Strengths:
  * Clearly a good idea to improve an already strong baseline.

Weaknesses:
  * Clarity is lacking.
    * In equation (1), is the generation of candidiate labels conditioned on the correct label?  or perhaps conditionally independent of the correct label given the instance?  the notation in equation (1) is not clear and the surrounding text is silent.  (later in equation (3), there's a latent variable that is jointly conditioned with the instance, but not the correct label; and strangely equation (3) apparently attempts to model the probability that the correct label is [not] in the candidate set, whereas in PLL this is assumed always in the candidate set.)
    * I find the discussion around equation (10) completely unintelligible, but possibly related to my previous concern listed above.
  * A theoretical correctness concern:
    * In Lemma 1 and Theorem 1, boundedness of $\mathcal{L}_{MAP}$ does not superficially appear to be a reasonable assumption, as it has a negative log in it, but this is presented without comment.
  * Some experimental concerns:
    * The transformation of classification datasets to PPL datasets uses the assumed generative model, which degrades from all the conclusions drawn from it.  Consequently, I'm not drawing any of the conclusions you are hoping for:
      * Table 1 does not convince me the proposed approach is better.
      * Table 3 does not convince me regularization is beneficial.
      * Figure 1 does not convince me the technique is not overly sensitive to choice of hyperparameter, except for Figure 1b.
      * **Note**: Table 2 *is* persuasive.
      * **Constructive feedback**:
        * Drop Table 1 entirely.
        * Populate Figure 1 with 3 datasets from Table 2.
        * Do the comparison in Table 3 on datasets from Table 2.

**Summary Of The Paper:**

Authors extend VALEN with an instance-dependent incorrect label process.

**Summary Of The Review:**

**Don't panic.**  This paper wants to be accepted you just have to address my concerns and my score will change.

Factors that are creating a low score right now are:
1.  a confusing exposition in section 3.2 [have someone with strong writing skills read this section and help you edit it]
2. an apparently disqualifying precondition in lemma 1 and theorem 1 [justify it or change it]
3. use of your own generative model for evaluation [drop all use of these datasets and only use datasets from Table 2].


(Updated score from 3->6)

---

> ### Author Response · Authors · 2022-11-14
> **Response 1/2 to Reviewer bU6i**
>
> Thanks very much for your insightful comments and suggestions. We have conducted corresponding experiments and revised our paper based on the comments of you.
>
> ***Clarity:***
>
> **Q.1 In equation (1), is the generation of candidate labels conditioned on the correct label?  or perhaps conditionally independent of the correct label given the instance? The notation in equation (1) is not clear and the surrounding text is silent. (later in equation (3), there's a latent variable that is jointly conditioned with the instance, but not the correct label, and strangely equation (3) apparently attempts to model the probability that the correct label is [not] in the candidate set, whereas in PLL this is assumed always in the candidate set.）**
>
> We have added more explanations above equation (1) and revised the corresponding section, which may alleviate your confusion. Given an instance $x$, its candidate labels $S$ consist of the correct label $y$ and the incorrect candidate labels $\bar{S}$, the generation of which are both only conditioned on the instance $x$. Our generation model builds upon the PLL assumption that the correct label is always in the candidate set. In the derivation of equation (1), we directly split the candidate label set $S$ into the correct label $y=j(j\in S)$ and the incorrect candidate labels $\bar{S}^j$,i.e., $p(S|x) = \sum_{j \in S} p(y=j, \bar{S}^j | x)=\sum_{j \in S} p(y=j | x)p(\bar{S}^j | x)$. Then, in the following of Section 3, we explicitly model $p(y | x)$ and $p(\bar{S} | x)$, respectively. With the above consideration, more details around equation (1), such as "consisting of the correct label $y_i$ and the incorrect candidate labels $\bar{S}_i^{y_i}$", "Our generation model builds upon the PLL assumption that the correct label is always in the candidate set, which allows for directly splitting $S_i$ into $y_i$ and $\bar{S}_i^{y_i}$", have been given.
>
>
> **Q.2 l find the discussion around equation (10) completely unintelligible, but possibly related to my previous concern listed above.**
>
> We have revised the discussions around equation (10) and rewritten the last paragraph of Section 3 to elaborate on the optimization related to equation (10). Specifically, on Page 5, the paragraph above equation (10) explains why we require equation (10), which dynamically estimates the parameters of Dirichlet Distribution $\lambda$. Due to the existence of the memorization effects of Neural Networks, a neural network at the earlier training stage tends to learn the true pattern rather than overfit the incorrect labels. Hence, we use the weighted average of $\lambda^{(r)}$ at the early training epoch $r$ and $\lambda^{(t)}$ output by $f$ at the present epoch $t$ as the present estimation of the parameters of Dirichlet Distribution at the epoch $t$, i.e, $\hat{\lambda}^{(t)}$.
>
> ***Theoretical Correctness Concern:***
>
> **Q.3 In Lemma 1 and Theorem 1, boundedness of $L_{MAP}$ does not superficially appear to be a reasonable assumption, as it has a negative log in it, but this is presented without comment.**
>
> We have added the corresponding comment above Lemma 1 on Page 7 to explain the boundedness of $L_{MAP}$. The approaches [1, 2] in other weakly supervised paradigms (Noisy Label Learning, Complementary Label Learning) and the SOTA approaches [3,4] in PLL make similar assumptions on their loss functions when they deduce the generalization error bound. Though $L_{MAP}$ has a negative log like $-a \log b (a>0, 0<b<1)$, it will not extend to infinity under the SGD optimization and neural networks in practice. Intuitively, for the given dataset $ \mathcal{D}$, the bound $M$ takes the value $\max_{i}L(f(x_i), S_i)$ and $L_{MAP}$ will be no more than $M$. Hence, the corresponding part has been revised, and Lemma 1 and Theorem 1 have been presented with the comment "Inspired by Liu & Tao (2015); Feng et al. (2020), we make the assumption on the boundness of $L_{MAP}$, since it does not extend to infinity in practice."
>
> [1] Liu, Tongliang, and Dacheng Tao. "Classification with noisy labels by importance reweighting." IEEE Transactions on pattern analysis and machine intelligence 38.3 (2015): 447-461.
>
> [2] Yu, Xiyu, et al. "Learning with biased complementary labels." Proceedings of the European conference on computer vision (ECCV). 2018.
>
> [3] Lv, Jiaqi, et al. "Progressive identification of true labels for partial-label learning." International Conference on Machine Learning. PMLR, 2020.
>
> [4] Feng, Lei, et al. "Provably consistent partial-label learning." Advances in Neural Information Processing Systems 33 (2020): 10948-10960.

---

> > ### Author Response · Authors · 2022-11-14
> > **Response 2/2 to Reviewer bU6i**
> >
> > ***Some experimental concerns:***
> >
> > **Q.4 The transformation of classification datasets to PLL datasets uses the assumed generative model, which degrades from all the conclusions drawn from it.**
> >
> > 1) We have moved the original Table 1, where we use the assumed model to generate instance-dependent partial labels, and corresponding discussions to Appendix A.9. To enhance the persuasion, we have employed the same strategy as [1,2]  to corrupt the benchmark dataset into partially labeled versions. Each incorrect label $j$ corresponding to an instance $x_i$ has a flipping probability $\xi_{i}^{j}$ set by using the confidence prediction of a clean neural network $f(x;\hat{\Theta})$ (trained with the original clean labels) with $\xi_{i}^{j}=\frac{f_j(x_i;\hat{\Theta})}{\max_{k\neq y_i}f_k(x_i;\hat{\Theta})}$, where $y_i$ is the correct label of $x_i$.
> > Then we compare our method with other SOTA approaches on them and discuss the experimental results as the substitution. From Table 1, we can overall see that IDGP significantly outperforms all comparing approaches on all benchmark datasets (except on Kuzushiji-MNIST where VALEN performs comparably against IDGP).
> >
> > [1] Xu, Ning, et al. "Instance-dependent partial label learning." Advances in Neural Information Processing Systems 34 (2021): 27119-27130.
> >
> > [2] Wu, Dong-Dong, Deng-Bao Wang, and Min-Ling Zhang. "Revisiting consistency regularization for deep partial label learning." International Conference on Machine Learning. PMLR, 2022.
> >
> > *Table 1 Comparison with the SOTA approaches*
> >
> > | **Datasets** | **MNIST**                 | **FMNIST**                | **KMNIST**                | **CIFAR10**               | **CIFAR100**              |
> > |:------------:|:-------------------------:|:-------------------------:|:-------------------------:|:-------------------------:|:-------------------------:|
> > | IDGP         |  **98.92 $\pm$ 0.05\%**     |  **91.48 $\pm$ 0.32\%**    |  **96.88 $\pm$ 0.17\%**     |  **87.57 $\pm$ 0.26\%**     | **64.59 $\pm$ 0.17\%** |
> > | PLCR         | 98.56 $\pm$ 0.08\% $\bullet$  | 90.10 $\pm$ 0.21\% $\bullet$  | 95.29 $\pm$ 0.21\% $\bullet$  | 86.37 $\pm$ 0.38\% $\bullet$  | 64.12 $\pm$ 0.23\% $\bullet$  |
> > | PICO         | 98.61 $\pm$ 0.12\% $\bullet$  | 88.41 $\pm$ 0.20\% $\bullet$  | 94.78 $\pm$ 0.19\% $\bullet$  | 86.16 $\pm$ 0.21\% $\bullet$  | 62.98 $\pm$ 0.38\% $\bullet$  |
> > | VALEN        | 98.72 $\pm$ 0.05\% $\bullet$  | 90.63 $\pm$ 0.30\% $\bullet$  | 96.19 $\pm$ 0.75\%          | 85.48 $\pm$ 0.62\% $\bullet$  | 62.96 $\pm$ 0.96\% $\bullet$  |
> > | CAVL         | 98.84 $\pm$ 0.05\% $\bullet$  | 87.94 $\pm$ 0.19\% $\bullet$  | 93.69 $\pm$ 0.28\% $\bullet$  | 59.67 $\pm$ 3.30\% $\bullet$  | 52.59 $\pm$ 1.01\% $\bullet$  |
> > | LWS          | 98.56 $\pm$ 0.06\% $\bullet$  | 88.99 $\pm$ 0.26\% $\bullet$  | 92.27 $\pm$ 1.03\% $\bullet$  | 37.49 $\pm$ 2.82\% $\bullet$  | 53.98 $\pm$ 0.99\% $\bullet$  |
> > | RC           | 98.41 $\pm$ 0.09\% $\bullet$  | 89.60 $\pm$ 0.19\% $\bullet$  | 93.78 $\pm$ 0.17\% $\bullet$  | 85.95 $\pm$ 0.40\% $\bullet$  | 63.41 $\pm$ 0.56\% $\bullet$  |
> > | CC           | 98.16 $\pm$ 0.14\% $\bullet$  | 89.86 $\pm$ 0.11\% $\bullet$  | 94.08 $\pm$ 0.35\% $\bullet$  | 79.96 $\pm$ 0.99\% $\bullet$  | 62.40 $\pm$ 0.84\% $\bullet$  |
> > | PRODEN       | 98.39 $\pm$ 0.10\% $\bullet$  | 89.79 $\pm$ 0.24\% $\bullet$  | 93.79 $\pm$ 0.24\% $\bullet$  | 86.04 $\pm$ 0.21\% $\bullet$  | 62.56 $\pm$ 1.49\% $\bullet$  |
> >
> > 2) We have populated Figure 1 with three real-world datasets (Lost, BirdSong and Yahoo!News) from Table 2. As illustrated in Table 3, IDGP achieves superior performance on all real-world datasets compared to IDGP-ML with the assistance of the prior information.
> >
> > 3) We have done the comparison in Table 3 on datasets from Table 2.
> >
> > *Table 3 Ablation study.*
> > | **Dataset**   | **IDGP**                  | **IDGP-ML**               |
> > |:-------------:|:-------------------------:|:-------------------------:|
> > | Lost          |  **77.02 $\pm$ 0.82 \%**  | 52.50 $\pm$ 3.20\% $\bullet$  |
> > | BirdSong      |  **74.23 $\pm$ 0.17 \%**  | 71.13 $\pm$ 0.45\% $\bullet$  |
> > | MSRCv2        |  **50.45 $\pm$ 0.47 \%**  | 43.04 $\pm$ 0.98\% $\bullet$  |
> > | Soccer Player |  **55.99 $\pm$ 0.28 \%**  | 49.56 $\pm$ 0.31\% $\bullet$  |
> > | Yahoo!News    |  **66.62 $\pm$ 0.19 \%**  | 51.48 $\pm$ 0.07\% $\bullet$  |
> >
> > The paper have been updated with the above results and discussions in Section 4.
> >
> > **Summary of revision**
> >
> > (1) Clarity: We have edited Section 3.2 with more details provided. Besides, we have rewritten the last paragraph of Section 3 to further clarify the optimization of our algorithm.
> >
> > (2) Theoretical Correctness Concern: We have cited the papers related to the precondition about the boundedness of $L_{MAP}$  in Lemma 1 and Theorem 1 and made the explanation for justification.
> >
> > (3) Some experimental concerns: We appreciate for your constructive feedbacks and have conducted the corresponding experiments and revised the experimental discussions.

---

> ### Author Response · Authors · 2022-11-28
> **Thank you again for your valuable time and efforts in improving this work**
>
> Dear Reviewer bU6i,
>
> Please feel free to ask us anything if there are any additional questions. We will immediately respond to them. We understand you are very busy and appreciate your time. Your feedback is valuable to us--we will be waiting for it.
>
> Thanks again!
>
> Authors

---

> ### Comment · Reviewer_bU6i · 2022-12-08
> **Update**
>
> Authors have addressed many of my concerns with substantive changes, hence my score is changing accordingly.

---

### Official Review · Reviewer_YnxC · 2022-10-23

**Confidence:** 4
**Correctness:** 4
**Technical Novelty And Significance:** 3
**Empirical Novelty And Significance:** 3
**Recommendation:** 8

**Clarity, Quality, Novelty And Reproducibility:**

The paper is well-written and easy to follow. The extensive empirical results on both synthetic and real-world datasets validate the effectiveness of the proposed method.  Since the approaches are not difficult, the reproducibility could be ensured.



**Strength And Weaknesses:**

Strengths:
1. This paper provides a new perspective on instance-dependent partial label learning.
2. This paper provides a theoretical analysis for the estimation error bound of the proposed empirical risk.
3. As far as I know, it is the first attempt to explicitly models the generation process of the correct labels and incorrect positive labels from different distributions.
4. The theoretical justifications and empirical validations are solid.

Weaknesses:
1. The authors should give more discussion about the results on real-world PLL datasets.
2. The authors should give more details the MAP optimization problem in Eq. (7)
3. “Then we perform Maximum A Posterior(MAP) estimation” should be revised as “Then we perform Maximum A Posterior (MAP) estimation”.



**Summary Of The Paper:**

This paper investigates the problem of partial label learning (PLL) on the instance-dependent scenario, which is a more realistic and practical setting. A new generative model is proposed that decomposes the generative process into two parts related to the ground-truth labels and other candidate labels, respectively. The distribution of the correct label and the incorrect label in the candidate label set of each training example are modeled explicitly by decoupled probability distributions Categorical distribution and Bernoulli distribution. MAP estimation is employed on the PLL training dataset to deduce a risk minimizer. Experiments on benchmark and real-world datasets validate the effectiveness of the proposed method.

**Summary Of The Review:**

Overall, this is a well-written paper and proposes a novel approach for instance-dependent partial label learning.

---

> ### Author Response · Authors · 2022-11-14
> **Response to Reviewer YnxC**
>
> We thank you for the critical comments and helpful suggestions. We have taken all these comments and suggestions into account, and have made major corrections in the revision.
>
> **1.The authors should give more discussion about the results on real-world PLL datasets.**
>
> We have further discussed the results on real-world PLL datasets in the revision. Specifically, we conduct the ablation study on real-world PLL datasets and add the sensitivity analysis for the real-world dataset Lost, which shows a more detailed performance of our approaches on real-world datasets.
>
> **2.The authors should give more details about the MAP optimization problem in Eq. (7)**
>
> We have given more details the MAP optimization problem in Eq. (7)  in the revision. To be specific, more derivations related to Eq.(7) are shown in Appendix A.1 and A.2.  And the optimization step about Eq.(7) at the epoch $t$ are rewritten in more detail in the last paragraph on Page 6.
>
> **3.“Then we perform Maximum A Posterior(MAP) estimation” should be revised as “Then we perform Maximum A Posterior (MAP) estimation"**
>
> We have corrected the mistake in the revision.

---

### Official Review · Reviewer_XV9C · 2022-10-24

**Confidence:** 5
**Correctness:** 4
**Technical Novelty And Significance:** 4
**Empirical Novelty And Significance:** 4
**Recommendation:** 8

**Clarity, Quality, Novelty And Reproducibility:**

The paper is simple to read, and the results are well presented. In this paper, the idea of decomposing and modeling the generation process is novel in PLL. The optimization procedure and objective are provided, and replicating the approach appears to be simple.

**Strength And Weaknesses:**

Strength:
1.  It makes sense that the paper decomposes and models the generation process of candidate labels with two separate processes. The generation process is reasonable and interesting, and it has the potential to benefit future research.

2.  This definition of the data generation process is mathematically clear. The reasoning that follows is standard and rigorous, including the derivation of the risk estimator and the statistical analysis of the generalization error bound, is standard and rigorous.  All of these contribute to the soundness of the paper.

3.  The proposed approach IDGP indicates the superiority when it is compared to the existing PLL methods in benchmark and real-world datasets.

Weakness:
1. In the paper, the authors respectively utilize Categorical and Bernoulli distributions to capture these two processes. More explanations can be provided on it to further clarify why the instance-dependent partial labels could be modeled by these two distributions.

2. The presentation is good, but it might be better to use visuals and figures to provide an additional explanation of the learning procedure.

3. The approach in Proposed Method includes two branch models, the main branch $f$ and the auxiliary branch $g$. The manually corrupted benchmark datasets in Experiments also use two sub-models $f$ and $g$. Overall, the former corresponds to the latter. However, the former $g$ appears to output two variables, $alpha$ and $beta$, whereas the latter $g$ appears to output only one. The difference inside should be explained in detail.

4. Some related works [1, 2] are suggested to be cited and discussed.
[1] Lyu, Gengyu, Yanan Wu, and Songhe Feng. "Deep Graph Matching for Partial Label Learning."
[2] He, Shuo, et al. "Partial Label Learning with Semantic Label Representations."


**Summary Of The Paper:**

The paper proposes an instance-dependent PLL approach by decomposing the generation process of candidate labels into two processes and explicitly modeling the processes using different probability distributions, which the risk estimator is built upon. The MAP technique is employed to create the final empirical risk estimator. Then, the generation error bound of it is proposed for it in the theoretical analysis. Finally, experiments on corrupted benchmark datasets and real-world datasets validate the effectiveness of the proposed approach.

**Summary Of The Review:**

This paper is intriguing and makes good contributions to Instance-Dependent Partial Label Learning.

---

> ### Author Response · Authors · 2022-11-14
> **Response to Reviewer XV9C**
>
> We'd like to thank you for your careful readings and helpful comments. We agree with you that the idea of decomposing and modeling the generation process is novel in PLL. We provide some responses and clarifications.
>
>
> **1.In the paper, the authors respectively utilize Categorical and Bernoulli distributions to capture these two processes. More explanations can be provided on it to further clarify why the instance-dependent partial labels could be modeled by these two distributions.**
>
> In Section 1 and Section 3.2, we mention two processes related to the annotator. One is that the annotator select one label from $c$ classes, which is similar to doing single-choice questions. The sampling process satisfies Categorical Distribution. Another is that the annotator is disturbed by multiple incorrect classes. At this time, the annotator is doing $|c|$ judgment questions. Each sampling process of incorrect labels satisfies Bernoulli Distribution.
>
> **2.The presentation is good, but it might be better to use visuals and figures to provide an additional explanation of the learning procedure.**
>
> The last paragraph of Section 3 has been rewritten to further clarify the learning procedure of our algorithm with more details. We expect it can alleviate the confusion.
>
>
> **3.The approach in Proposed Method includes two branch models, the main branch $f$ and the auxiliary branch $g$. The manually corrupted benchmark datasets in Experiments also use two sub-models $f$ and $g$. Overall, the former corresponds to the latter. However, the former $g$ appears to output two variables, $\alpha$ and $\beta$, whereas the latter $g$ appears to output only one. The difference inside should be explained in detail.**
>
> To simplify the generation procedure in the experiment, the latter $g$ only outputs $\alpha$ and uses the $sigmoid$ function for activation. The former $g$ outputs $\alpha, \beta$ and $z_i^j$ is calculated by $z_i^j=\frac{\alpha_i^j}{\alpha_i^j + \beta_i^j}$. The latter $g$ calculates $z_i^j=\frac{1}{1+e^{-g(x)}}$, which amounts to the case that $\alpha=1, \beta=e^{-g(x)}$.
>
>
> **4.Some related works [1, 2] are suggested to be cited and discussed. [1] Lyu, Gengyu, Yanan Wu, and Songhe Feng. "Deep Graph Matching for Partial Label Learning." [2] He, Shuo, et al. "Partial Label Learning with Semantic Label Representations."**
>
>
> We have cited and discussed in the section Related Work in the revision.

---

### Official Review · Reviewer_Zc1G · 2022-10-25

**Confidence:** 4
**Correctness:** 4
**Technical Novelty And Significance:** 4
**Empirical Novelty And Significance:** 4
**Recommendation:** 8

**Clarity, Quality, Novelty And Reproducibility:**

The paper is well organized and theoretically sound, indicating its good clarity and high quality. It is novel to decompose the generation process and explicitly model it using two probabilistic distributions. The overall proposed approach is easy to understand, so I believe that its reproducibility is assured.

**Strength And Weaknesses:**

Pros:
1.	The proposed generation process is inspiring and offers a new perspective for me on Instance-Dependent Partial Label Learning. The generation process is distinct from traditional Multi-class Learning and Multi-Label Learning, but it also shares a few similarities, which is significant for understanding this problem.
2.	The paper is well-organized and logical. It begins by explicitly modeling the process with decoupled probabilistic distributions, then forms the optimization objective using a MAP technique, and finally theoretically analyzes the estimation error of the optimization objective.
3.	The experimental results are superior to the baselines, demonstrating the effectiveness of the proposed approach.

Cons and Qs:
1. Since the optimization is based on the proposed generation model, the log-likelihood
loss function in Eq.(4), which is derived from Eq.(1)~(3), could be more detailed. On the other hand, it is also used for ablation studies. Hence, more explanations need to be made to convince me.
2. Could you explain whether the proposed process is adaptable to a uniform generation process of candidate labels or the relationship between them?
3. I am curious about why Eq. (6) holds. Because the derivation is not in the appendix, a detailed explanation would be greatly appreciated.
4. I'm not sure how to estimate the parameters $\lambda$, $\alpha$, $\beta$, $\theta$ and $z$. Eq.(8) and Eq.(9) use $\lambda$, $\alpha$, $\beta$ to calculate $\hat{\alpha}$, $\hat{\beta}$, while Eq.(10) and Eq.(11) replace $\lambda$, $\alpha$, $\beta$ with $\hat{\lambda}$, $\hat{\alpha}$, $\hat{\beta}$.


**Summary Of The Paper:**

This paper aims to address the challenging problem of Instance-Dependent Partial Label Learning, which is caused by overfitting on candidate labels, by rethinking the generation process of instance-dependent partial labels. In detail, the authors consider a two-step decomposition-based process. Categorical Distribution generates the correct label for a given instance, and Bernoulli Distribution samples the incorrect labels. In this way, they explicitly model the generation process. Then, to form their optimization objective, Maximum A Posterior (MAP) is performed, with Dirichlet Distribution and Beta Distribution introduced as prior distributions.

**Summary Of The Review:**

Overall, the paper proposes an insightful and novel approach to dealing with the instance-dependent PLL, and the experimental results demonstrate its effectiveness. As a result, I vote for acceptance.

---

> ### Author Response · Authors · 2022-11-14
> **Response to Zc1G**
>
> We'd like to thank you for your careful readings and helpful comments. We agree with you that the proposed generation process is inspiring and offers a new perspective on Instance-Dependent Partial Label Learning.. We provide some responses and clarifications.
>
>
> **Q.1 Since the optimization is based on the proposed generation model, the log-likelihood loss function in Eq.(4), which is derived from Eq.(1)~(3), could be more detailed. On the other hand, it is also used for ablation studies. Hence, more explanations need to be made to convince me.**
>
> The derivation of Eq.(4) is not difficult. We provide it in the appendix.
> $$
> L_{ML}=-\sum_{i=1}^{n} {\log} p(S_{i}| x_i, \theta_i, z_i)=-\sum_{i=1}^{n} {\log} \sum_{j\in S_i} p(y_i = j | x_i)p(\bar{S}_i^j | x_i)
> $$
>
> $$
> =-\sum_{i=1}^{n} {\log} \sum_{j \in S_i} (Cat(l_i | x_i , \theta_i) \prod_{k=1}^{c}Ber(\bar{s_i^k|z_i^k}) | y_i=j)
> $$
>
> $$
> =-\sum_{i=1}^{n} {\log} \sum_{j \in S_i} ( \prod_{k=1}^{c} (\theta_i^k)^{l_i^k} \prod_{k=1}^{c} (z_i^k)^{\bar{s_i^k}}(1-z_i^k)^{1-\bar{s_i^k}}| y_i=j)
> $$
>
> $$
> =-\sum_{i=1}^{n} {\log} \sum_{j \in S_i} \theta_i^j \prod_{k\in \bar{S_i^j}} z_i^k \prod_{k \notin \bar{S_i^j} } (1 - z_i^k)
> $$
>
>
> **Q.2 Could you explain whether the proposed process is adaptable to a uniform generation process of candidate labels or the relationship between them?**
>
> The proposed process can accommodate the uniform generation process of candidate labels by setting parameters of Bernoulli distribution to a constant $p$, which means the flipping probability. In this case, we can degenerate $L_{ML}, L_{reg}$ and $L_{MAP}$ to the following form.
> $$
> L_{ML} = -\sum_{i=1}^{n} {\log} \sum_{j \in S_i} \theta_i^j \prod_{k\in \bar{S_i^j}} z_i^k \prod_{k \notin \bar{S_i^j} } (1 - z_i^k)= -\sum_{i=1}^{n} {\log} \sum_{j \in S_i} \theta_i^j  (1-p)^{c+1-|S_i|} p^{|S|-1}
> $$
>
> $$
> = -\sum_{i=1}^{n}  {\log} (1-p)^{c+1-|S_i|} p^{|S|-1}  \sum_{j \in S_i} \theta_i^j
> $$
>
> $$
> = -\sum_{i=1}^{n} {\log}  \sum_{j \in S_i} \theta_i^j - \sum_{i=1}^{n}  {\log} (1-p)^{c+1-|S_i|} p^{|S|-1}
> $$
>
> The second term is a constant, so the final $L_{ML}$ is formulated as
>
> $$
> L_{ML} = -\sum_{i=1}^{n} {\log}  \sum_{j \in S_i} \theta_i^j
> $$
>
> Due to the parameters of Bernoulli Distribution are constant, we do not need Beta Distribution. Hence,
> $$
> L_{reg} = -\sum_{i=1}^{n} \sum_{j=1}^{c} (\lambda_i^j -1 ) {\log} \theta^i_j
> $$
>
> Finally,
> $$
> L_{MAP} = L_{ML} + L_{reg} = -\sum_{i=1}^{n} {\log}  \sum_{j \in S_i} \theta_i^j -\sum_{i=1}^{n} \sum_{j=1}^{c} (\lambda_i^j -1 ) {\log} \theta^i_j
> $$
>
>
> **Q.3 I am curious about why Eq. (6) holds. Because the derivation is not in the appendix, a detailed explanation would be greatly appreciated.**
>
> The derivation of Eq.(6) has been provided in the appendix.
> $$
> L_{reg} = - \sum_{i=1}^{n} {\log} p(\theta_i | x_i) + {\log} p(z_i | x_i)
> $$
>
> $$
> =- \sum_{i=1}^{n} {\log} \frac{\Gamma(\sum_{j=1}^{c}\lambda_{i}^j)}{\prod_{j=1}^{c}\Gamma(\lambda_i^j)}\prod_{j=1}^{c} (\theta_i^j)^{\lambda_i^j-1} + {\log} \prod_{j=1}^{c} \frac{\Gamma (\alpha_i^j + \beta_i^j)}{ \Gamma (\alpha_i^j) \Gamma (\beta_i^j)} (z_i^j)^{\alpha_i^j -1} (1-z_i^j)^{\beta_i^j - 1}
> $$
>
> $$
> =- \sum_{i=1}^{n} {\log} \prod_{j=1}^{c}(\theta_i^j)^{\lambda_i^j -1} + {\log}\prod_{j=1}^{c} (z_i^j)^{\alpha_i^j - 1} (1-z_i^j)^{\beta_i^j - 1} + C
> $$
>
> $$
> =- \sum_{i=1}^{n} \sum_{j=1}^{c} (\lambda_i^j -1) {\log} (\theta_i^j)  + (\alpha_i^j -1) {\log} (z_i^j) + (\beta_i^j - 1)  {\log} (1 - z_i^j) + C
> $$
>
> $C=- \sum_{i=1}^{n} {\log} \frac{\Gamma(\sum_{j=1}^{c}\lambda_{i}^j)}{\prod_{j=1}^{c}\Gamma(\lambda_i^j)} +  {\log} \prod_{j=1}^{c} \frac{\Gamma (\alpha_i^j + \beta_i^j)}{ \Gamma (\alpha_i^j) \Gamma (\beta_i^j)}$. Due to that we later use $\hat{\lambda}$, $\hat{\alpha}$ and $\hat{\beta}$ in Eq.(10) and Eq.(11) to replace $\lambda$, $\alpha$ and $\beta$ in Eq.(7), which are fixed to be constants, C is a constant and ignored. Here, the derivation of Eq.(6) has been finished.
>
>
> **Q.4 I'm not sure how to estimate the parameters $\lambda$, $\alpha$, $\beta$, $\theta$ and $z$. Eq.(8) and Eq.(9) use $\lambda$, $\alpha$, $\beta$ to calculate $\hat{\alpha}$, $\hat{\beta}$, while Eq.(10) and Eq.(11) replace $\lambda$, $\alpha$, $\beta$ with $\lambda$, $\hat{\alpha}$, $\hat{\beta}$.**
>
> We have explained the estimation process and added more details to further clarify it on Page 5.  Besides, we have rewritten the last paragraph of Section 3 to further clarify the optimization of our algorithm.  $f$ outputs $\lambda$ while $g$ outputs $\alpha$ and $\beta$. According to Eq.(8) and Eq.(9), we estimate $\theta$ and $z$ with $\hat{\theta}$ and $\hat{z}$ for the optimization in Eq.(7). Next, to introduce prior information, we use $\hat{\lambda}$, $\hat{\alpha}$ and $\hat{\beta}$ in Eq.(10) and Eq.(11) to replace $\lambda$, $\alpha$ and $\beta$ in Eq.(7). Note that $\hat{\theta}$ and $\hat{z}$ are variables, of which we calculate the gradient to perform backpropagation, while $\hat{\lambda}$, $\hat{\alpha}$ and $\hat{\beta}$ are constants.

---

### Decision · Program_Chairs · 2023-01-20

**Decision:**

Accept: notable-top-25%

**Justification For Why Not Higher Score:**

The score of this paper is "Accept (spotlight)" now. The concern that makes it not be selected for oral presentation is that this paper's contributions are focused on partial label learning. Although the core idea is interesting, it may be limited in other weakly-supervised learning settings since it relies on specific prior distributions. Besides, more clarifications of method descriptions and theoretical analysis are still needed.


**Justification For Why Not Lower Score:**

This paper makes solid contributions to partial label learning. The core idea of rethinking the generation process is interesting and can inspire follow-up research in partial label learning. Therefore, AC and reviewers unanimously think that this paper should be accepted and highlighted.

**Metareview: Summary, Strengths And Weaknesses:**

This paper targets a typical weakly-supervised learning problem named partial label learning. A novel method is proposed by rethinking the generation process of instance-dependent partial labels. Specifically, a two-step decomposition-based process is considered, where two steps are related to the ground-truth labels and other candidate labels respectively. The MAP technique is used to create the empirical risk estimator. Both theoretical analysis and empirical verifications are provided to confirm the effectiveness of the proposed method.

The strengths of this paper include: (1) It is insightful to handle the problem of partial label learning from the generation process. The rationality of the proposed generation process in this paper is well justified. Besides, this idea can inspire follow-up research in partial label learning. (2) Both theoretical justifications and empirical validations are solid. This paper gives a detailed analysis of the estimation error bound of the predictive model. Additionally, comprehensive experiments indicate the superiority of the proposed method when it is in various benchmarks. (3) The presentation of the paper is logical. The weaknesses of this paper mainly lie in that it needs more clarifications of method descriptions and theoretical analysis.

In the rebuttal process, the authors provide detailed responses to the concerns of reviewers. All reviewers then vote for accepting this paper. AC checks all reviewing activities and agrees with reviewers that this paper makes solid contributions to partial label learning.

**Note From Pc:**

if the above contains the word "oral" or "spotlight" please see: "oral" presentation means -> notable-top-5% and "spotlight" means -> notable-top-25%. As stated in our emails, we are disassociating presentation type from AC recommendations

**Summary Of Ac-Reviewer Meeting:**

N/A